# Elucidating Bile Acid Tolerance in *Saccharomyces cerevisiae*: Effects on Sterol Biosynthesis and Transport Protein Expression

**DOI:** 10.3390/foods13213405

**Published:** 2024-10-25

**Authors:** Miao Zheng, Qi Su, Haoqing Wu, Chenggang Cai, Le Thanh Ninh, Haiying Cai

**Affiliations:** 1School of Biological and Chemical Engineering, Zhejiang University of Science & Technology, Hangzhou 310023, China; mmiaozheng942@gmail.com (M.Z.); suqi99-@outlook.com (Q.S.); whq1256939126@outlook.com (H.W.); ccg0516@sina.com (C.C.); 2Department of Food Science and Engineering, National University of Singapore, Singapore 117542, Singapore; itninh90@gmail.com; 3College of Biosystems Engineering and Food Science, Zhejiang University, Hangzhou 310058, China

**Keywords:** *Saccharomyces cerevisiae*, ATP-binding cassette transporters, bile acids, tolerance mechanisms

## Abstract

The tolerance of *Saccharomyces cerevisiae* to high concentrations of bile acids is intricately linked to its potential as a probiotic. While the survival of yeast under high concentrations of bile acids has been demonstrated, the specific mechanisms of tolerance remain inadequately elucidated. This study aims to elucidate the tolerance mechanisms of *S. cerevisiae* CEN.PK2-1C under conditions of elevated bile acid concentrations. Through growth curve analyses and scanning electron microscopy (SEM), we examined the impact of high bile acid concentrations on yeast growth and cellular morphology. Additionally, transcriptomic sequencing and molecular docking analyses were employed to explore differentially expressed genes under high bile acid conditions, with particular emphasis on ATP-binding cassette (ABC) transporters and steroid hormone biosynthesis. Our findings indicate that high concentrations of bile acids induce significant alterations in the sterol synthesis pathway and transporter protein expression in *S. cerevisiae*. These alterations primarily function to regulate sterol synthesis pathways to maintain cellular structure and sustain growth, while enhanced expression of transport proteins improves tolerance to elevated bile acid levels. This study elucidates the tolerance mechanisms of *S. cerevisiae* under high bile acid conditions and provides a theoretical foundation for optimizing fermentation processes and process control.

## 1. Introduction

*Saccharomyces cerevisiae,* as a traditional fermentation species, serves as a crucial model organism for the study of eukaryotic cell growth, development, and apoptosis [1]. With its well-characterized genetic background and defined metabolic pathways, this yeast has been extensively utilized in the efficient synthesis of energy materials and biopharmaceutical products over time, earning it the designation of the safest and most systematic cellular factory. The products synthesized or metabolized by *S. cerevisiae* have found widespread applications in the food and pharmaceutical industries [2]. In recent years, species such as *S. cerevisiae*, *Issatchenkia orientalis*, and *Yarrowia lipolytica* have been developed as cell factories for the production of high-value natural products, demonstrating significant application potential [3]. For instance, Zhang et al. [4] achieved the de novo synthesis of the snake venom protein scaffold, snake venom protein F, and its oxidative derivatives using *S. cerevisiae*, thus addressing the gap in heterologous synthesis of ester terpenoids in this yeast. Zhang et al. [5] through the engineered modification of the pyruvate dehydrogenase (PDH) bypass, knockout of the CIT2 gene, and heterologous expression of *ScACSL641P*, attained a yield of 2.23 g/L of the monoterpene compound limonene in fed-batch shake flask fermentation. Rendulić et al. [6] successfully enhanced the flux of the oxidative tricarboxylic acid cycle by targeting mitochondrial uptake of pyruvate and cytoplasmic intermediates as well as modulating the succinate dehydrogenase complex, resulting in a 27% increase in succinate production. Additionally, the application of CRISPR-Cas9 gene editing technology in Saccharomyces cerevisiae cell factories has significantly accelerated the development of microbial engineering, reducing labor and time costs and deepening the understanding of yeast genetics and physiology [7]. These advancements provide a solid foundation for the further development and application of *S. cerevisiae* as an industrial cell factory.

At the same time, there has been increasing interest in the probiotic potential of yeast. Certain yeast strains can produce metabolites beneficial to gut health, such as short-chain fatty acids (SCFAs), which serve multiple physiological functions, including the promotion of intestinal epithelial cell maturation and the enhancement of immune function [8]. Moreover, specific strains, such as *Saccharomyces boulardii*, have demonstrated significant efficacy in the treatment and prevention of various gastrointestinal disorders, including inflammatory bowel disease, irritable bowel syndrome, antibiotic-associated diarrhea, and infectious diarrhea [9].

Bile acids, essential metabolites derived from cholesterol, are fundamental to numerous physiological processes, including digestion, lipid and glucose metabolism, inflammation, and immune responses in both humans and animals. As the primary active components of bile, they are synthesized in the liver and stored in the gallbladder. During digestion, bile acids are released into the small intestine, facilitating fat digestion and the absorption of fat-soluble vitamins. By modulating fatty acid biosynthesis, insulin signaling, and the AMPK signaling pathway, bile acids influence total cholesterol, low-density lipoprotein cholesterol, triglycerides, and hepatic cholesterol levels. Supplementation with bile acids has been shown to enhance lipid metabolism, likely by downregulating lipogenic genes and protease expression while promoting lipolysis [10]. Studies indicate that adding bile acids as a feed additive significantly improves feed conversion ratio (FCR) and breast muscle development in broilers fed a high-fat diet, primarily through the FXR-mediated IGF2 pathway [11]. Additionally, research by Liu et al. [12] demonstrated that exogenous bile acid supplementation enhances fat utilization from feed, combats fatty liver, boosts immune function, promotes bile secretion, maintains biliary patency, and exhibits bactericidal effects. These findings highlight the pivotal role of bile acids in regulating lipid metabolism in livestock, effectively promoting growth and significantly enhancing product quality [11,13].

Despite their essential physiological roles, the high concentration and acidic nature of bile acids can be harmful to many probiotic microorganisms. Therefore, the bile acid tolerance of yeast is crucial for its colonization and functional efficacy in the gut, a critical prerequisite for its probiotic activity in the gastrointestinal tract. This tolerance has significant implications for the development and application of probiotics. Bile acids have attracted considerable attention for their anti-inflammatory properties, metabolic regulatory functions, immunosuppressive effects, and anticancer activities. While their mechanisms of action in animals and humans have been extensively studied, their effects on probiotic microorganisms remain relatively underexplored. This study aims to investigate the impact of bile acid stress on the growth and tolerance of *S. cerevisiae*. Through transcriptomic analysis and molecular docking, we seek to elucidate the underlying mechanisms, providing a theoretical foundation for the application of bile acids in yeast-based probiotics.

## 2. Materials and Methods

### 2.1. Strains, Media, and Reagents

*Saccharomyces cerevisiae* (CEN.PK2-1C, referred to as Y) was cultured in yeast extract peptone dextrose (YPD) medium, which consisted of 20 g/L glucose, 10 g/L yeast extract, and 20 g/L peptone. For solid YPD medium, 20 g/L agar was added. The YPD medium was obtained from Nanjing Jiancheng Bioengineering Institute. Deoxycholic acid (DCA), lithocholic acid (LCA), chenodeoxycholic acid (CDCA), ursodeoxycholic acid (UDCA), and cholic acid (CA) were sourced from Shanghai Aladdin Biochemical Technology.

### 2.2. Optical Density (OD) Measurement

The *S. cerevisiae* (Y) strain was retrieved from a −80 °C glycerol stock and inoculated onto YPD solid medium, followed by incubation at 28 °C for 4 or 5 days. After activation, the culture was transferred into 15 mL of fresh YPD medium and incubated at 28 °C with shaking at 200 rpm. The cells were then harvested by centrifugation at 8000 rpm for 10 min at room temperature. The resulting cell pellet was resuspended in fresh medium containing varying concentrations of bile acids—deoxycholic acid (DCA), lithocholic acid (LCA), chenodeoxycholic acid (CDCA), ursodeoxycholic acid (UDCA), and cholic acid (CA)—at 0, 0.2, 1, 2, and 5 mg/mL. The cultures were incubated at 28 °C with shaking at 200 rpm, and the optical density (OD) at 600 nm was recorded every 12 h.

### 2.3. Scanning Electron Microscopy (SEM)

Samples from different groups were collected, and after centrifugation, the supernatants were discarded. The harvested *S. cerevisiae* cells were fixed in 3% glutaraldehyde solution for 24 h. Following fixation, the cells were dehydrated through a graded ethanol series (ranging from 30% to 100% ethanol) and subsequently dried in an oven at 30 °C. Finally, the cells were examined by scanning electron microscopy (SEM) (Hitachi, SU1510, Tokyo, Japan) to observe their morphology [14].

### 2.4. Sample Preparation, Collection, and Transcriptomic Sequencing Analysis of Saccharomyces cerevisiae Cells

*Saccharomyces cerevisiae* was first inoculated into YPD medium and incubated at 28 °C for 48 h. A single colony was then selected and transferred into 50 mL of medium containing 4 mg/mL deoxycholic acid (DCA) as well as into a control medium without bile acids. The cultures were incubated at 28 °C with shaking at 200 rpm until they reached the logarithmic growth phase. The cells were then harvested, washed with sterile water, and frozen in dry ice for preservation.

RNA quality assessment and transcriptomic analysis were performed by Shanghai Majorbio Bio-pharm Technology Co., Ltd. (Shanghai, China). The sample processing involved the following steps: total RNA was extracted, and its concentration and purity were measured using a Nanodrop 2000 (Thermo Fisher Scientific, Waltham, MA, USA). RNA integrity was assessed by agarose gel electrophoresis, and RNA Quality Number (RQN) values were determined using the Agilent 5300 (Agilent, Santa Clara, CA, USA). For library construction, the RNA sample requirements included a total amount of 1 μg, a concentration of ≥30 ng/μL, an RQN > 6.5, and an OD260/280 ratio between 1.8 and 2.2. Magnetic beads with oligo(dT) were employed to bind the polyA tails, facilitating the isolation of mRNA from the total RNA for transcriptomic analysis. Fragmentation buffer was subsequently added to the enriched mRNA to randomly fragment it into approximately 300 bp pieces. Using reverse transcriptase and random primers, first-strand cDNA was synthesized from the mRNA template, followed by the synthesis of the second strand to form a stable double-stranded cDNA structure. End Repair Mix was then added to the double-stranded cDNA to create blunt ends. Subsequently, an A base was incorporated at the 3′ ends to facilitate the addition of adapter sequences. The adapter-ligated products were purified using the QIAquick PCR purification kit (Qiagen, Hilden, Germany), and PCR amplification was conducted to generate the final library. The transcriptome sequencing process was carried out as follows: high-throughput RNA sequencing was performed using the Illumina sequencing platform. The base quality and nucleotide content (A, T, G, C) of the raw sequencing reads (5′→3′) were assessed using Fastp software (version 0.23.0-, Shanghai, China). Reads that did not meet quality standards were discarded, and only high-quality reads (clean reads) were retained for further analysis.

### 2.5. Functional Annotation and Enrichment Analysis of Differentially Expressed Genes

Gene sequence alignment was conducted using Hisat2 software (version2.1.0, Johns Hopkins University, Baltimore, MD, USA). Functional annotation of the expressed genes was performed using several databases, including Clusters of Orthologous Groups of Proteins (COG), Non-Redundant Protein Sequence Database (NR), Gene Ontology (GO), KEGG, Swiss-Prot Protein Sequence Database (Swiss-Prot), and the Protein Families Database (Pfam). Differential expression analysis was executed using DESeq2 (version 1.24.0), DEGseq (version 1.38.0), and edgeR software (version 3.24.3). Differentially expressed genes (DEGs) were identified based on the criteria |log₂ FC| ≥ 1 (where FC denotes the fold change in gene expression between the experimental and control groups) and *p* ≤ 0.05. KEGG pathway enrichment analysis of gene sets was performed using the Majorbio Cloud Analysis Platform (https://cloud.majorbio.com/), with pathways considered significantly enriched when *p* < 0.05 [15].

### 2.6. Statistical Analysis

Data (*n* = 3) were analyzed and processed using Origin 2021 software (OriginLab Co., Northampton, MA, USA), and graphical representations were created with GraphPad Prism 10 software (Moutlsky, 2023, San Diego, CA, USA). Experimental data are presented as means ± standard deviations. Comparisons between data (*n* = 3) sets were conducted using one-way analysis of variance (ANOVA), with statistical significance determined at *p* < 0.05.

## 3. Results and Discussion

### 3.1. Effects of Bile Acids on the Growth of Saccharomyces cerevisiae

Different types of bile acids exhibit varying effects on the growth of *S. cerevisiae*, with their overall inhibitory impact being relatively mild and, in some cases, even exhibiting a stimulatory effect. As illustrated in Figure 1, LCA shows a concentration-dependent promotion of yeast growth. Long-term fermentation studies reveal that bile acids such as DCA, CDCA, UDCA, and CA each have differing influences on *S. cerevisiae* growth. The observed increases in optical density (OD) values in cultures supplemented with bile acids suggest a potential role for these compounds in promoting yeast growth. As steroid metabolites, bile acids may enhance yeast growth by modulating various metabolic processes. The differential effects of bile acids on yeast growth are likely related to their solubility, molecular structure, and the number and position of hydroxyl groups. The growth-promoting effect of LCA can be attributed to its low polarity and smaller molecular weight, owing to its single hydroxyl group, which facilitates its penetration into yeast cells and subsequently influences a range of biological processes.

### 3.2. Effects of Bile Acids on the Cellular Structure of Saccharomyces cerevisiae

To investigate the effects of bile acids on *S. cerevisiae*, scanning electron microscopy (SEM) was utilized to examine the cellular structure of the yeast (Figure 2). Compared to the control group, the yeast cells in the experimental groups exhibited reduced surface smoothness. Morphologically, the effects of deoxycholic acid (DCA), cholic acid (CA), chenodeoxycholic acid (CDCA), and ursodeoxycholic acid (UDCA) on yeast cell structure were minimal, likely due to the lower toxicity of these bile acids. Studies have shown that lithocholic acid is more toxic than other bile acids and prolonged intake can lead to hepatobiliary injury [16,17]; electron microscopy results confirmed that LCA caused the most pronounced changes in yeast cell structure. Nevertheless, based on the optical density (OD) values measured during the experiment, it is hypothesized that the presence of LCA in the medium induces a stress response in *S. cerevisiae*. Under high LCA concentration stress, yeast may activate various stress mechanisms to maintain normal growth. This stress response indicates that despite the significant effects of LCA on cellular structure, *S. cerevisiae* can adapt to this stress, thereby sustaining its survival and proliferation.

### 3.3. Transcriptomic Analysis of Saccharomyces cerevisiae Treated with High Concentrations of DCA

To further investigate the impact of bile acids on *S. cerevisiae*, transcriptomic sequencing and differential expression gene (DEG) analysis were performed on yeast cells treated with 4 mg/mL deoxycholic acid (DCA) (YD) compared to the control group (Y). A total of 39.98 Gb of clean data were obtained, with each sample yielding over 6.11 Gb and a Q30 base percentage exceeding 95.76%. After filtering, the number of clean reads in the treatment and control groups were 44,972,743 and 44,419,482, respectively. Based on gene expression changes in DCA-treated yeast cells, a total of 363 DEGs were identified, with 227 DEGs being upregulated and 136 DEGs downregulated. DEG expression patterns are illustrated in Figure 3b, which shows significant differences in gene expression between the two groups.

The KEGG pathway and Gene Ontology (GO) enrichment analyses of differentially expressed genes (DEGs) in *S. cerevisiae* cells treated with deoxycholic acid (DCA) are illustrated in Figure 4. GO analysis annotated the reference transcriptome, with 1108 genes associated with cellular components, 883 genes related to molecular functions, and 447 genes involved in biological processes (Figure 4a). In the cellular component category, genes were primarily linked to cell parts, organelles, organelle parts, and protein complexes. In the molecular function category, genes were predominantly associated with binding and catalytic activities. For biological processes, genes were chiefly involved in cellular processes, metabolic processes, and the organization or biogenesis of cellular components. To further elucidate the functions of DEGs, KEGG functional annotation analysis was conducted for the DEGs between the control and DCA-treated groups (Figure 4b). The DEGs were categorized into pathways such as metabolism, genetic information processing, environmental information processing, cellular processes, and organic systems.

The significantly enriched Gene Ontology (GO) annotation analysis is depicted in Figure 4c. The richness factor represents the degree of enrichment, with larger points indicating a greater number of transcripts associated with each GO term, and the color of the points reflecting different ranges of adjusted *p*-values. The analysis reveals that the GO annotations are predominantly concentrated on translation processes, including cytoplasmic translation, translation termination, the degradation of cellular protein complexes, and the breakdown of protein complexes. KEGG pathway enrichment analysis of the differentially expressed genes (DEGs) (Figure 4d) shows that these DEGs are mainly involved in ribosomal processes, steroid synthesis, transport proteins, and carbohydrate metabolism. Notably, DEGs from DCA-treated *S. cerevisiae* cells displayed significant annotations in steroid biosynthesis, indicating that genes related to steroid synthesis are differentially expressed under DCA treatment. These findings suggest that bile acids influence various aspects of protein synthesis and translation, energy metabolism, and cell division in *S. cerevisiae*. The addition of steroid compounds results in notable changes in yeast growth metabolism, with the most pronounced effects observed in ribosomal translation and steroid biosynthesis. However, the specific regulatory mechanisms underlying these processes warrant further investigation.

### 3.4. Potential Mechanisms of DCA Tolerance in Saccharomyces cerevisiae

Analysis of significantly differentially expressed genes (DEGs) revealed that transport proteins such as Pdr5p, Pdr10p, Pdr16p, Yor1p, and Snq2p exhibit substantial variations in expression, and these proteins are closely associated with the transport of ergosterol and bile acids [18]. This variation is intricately linked to the addition of bile acids. Transcriptomic data suggest that bile acids may enter *S. cerevisiae* cells through these transport proteins, influencing growth and metabolism. In response to stress, the expression of transport proteins, particularly *PDR5*, is significantly upregulated to expel accumulated toxins from the metabolic process, thereby maintaining cellular homeostasis [19]. Additionally, ergosterol plays a crucial role in yeast growth and adaptation [20]. In the presence of exogenous steroid compounds such as bile acids, yeast cells increase ergosterol content in the membrane to preserve membrane integrity. Consequently, proteins involved in the ergosterol biosynthesis pathway are also significantly upregulated (Figure 5, Table 1). Transport proteins like Prd5p are pivotal in this process, enhancing the expulsion of steroids to help the cell cope with high concentrations of steroid metabolites and regulating membrane permeability, which affects nutrient absorption and utilization.

Ergosterol is often referred to as a “fungal hormone” due to its crucial role in stimulating fungal growth and proliferation, significantly influencing the growth and morphological regulation of *S. cerevisiae* [21,22]. Research also indicates that in weak organic acid environments, the transcription levels of transport proteins such as Yor1p and Prd5p are upregulated, suggesting these proteins may aid yeast survival under acidic conditions [23]. Overexpression of *YOR1* has been shown to enhance yeast resistance to various antibiotics and improve adaptability to external stress [24]. Similarly, the overexpression of the PDR family of transporters can increase yeast resistance to steroid metabolites, primarily through the extrusion pump mechanism, which removes accumulated steroids or toxic metabolites from the cell. During fermentation, the interaction between ergosterol and transport proteins is crucial for stress adaptation. When exposed to exogenous steroid compounds, *S. cerevisiae* increases ergosterol content in the cell membrane to prevent membrane damage and maintain membrane permeability, thus supporting cell growth and development. Moreover, ergosterol regulates the activity of membrane-bound ATPases, influencing nutrient absorption and utilization, which further promotes yeast growth [25,26]. Davies et al. demonstrated that under hypoxic conditions, *S. cerevisiae* activates the steroid biosynthesis pathway to facilitate its growth [27]. Overall, bile acids enhance *S. cerevisiae*’s tolerance to steroid compounds by stimulating a high expression of transport proteins such as Pdr5p and Snq2p. The Pdr16p protein plays a key role in regulating the synthesis and transport of intracellular lipids, thereby altering membrane lipid composition, affecting membrane permeability, and preventing drug entry into the cell [28]. In *S. cerevisiae*, *PDR16* primarily regulates membrane permeability, while *PDR5* and *SNQ2* facilitate tolerance to high concentrations of steroid metabolites by extruding accumulated steroids. While the role of PDR family transport proteins in steroid resistance is well-documented, the specific regulatory mechanisms and their dual role in promoting cell growth and metabolic adaptation warrant further investigation, especially regarding how these proteins coordinate cellular functions under various stress conditions.

Interaction between transporter and ergosterol biosynthesis in *S. cerevisiae* is stimulated by deoxycholic acid. The genes highlighted in red are significantly upregulated, including transport proteins and enzymes required for ergosterol biosynthesis.

Further research has elucidated that the steroid biosynthesis pathway in *S. cerevisiae* is essential for maintaining cell growth and development through the production of ergosterol [29]. This biosynthetic process is intricately regulated by cell cycle factors. Steroid metabolites impact not only steroid synthesis and transport by activating transport proteins but also modulate intracellular transcription and translation processes. In experiments involving bile acid supplementation, key genes such as *ERG1*, *ERG3*, *ERG4*, *ERG5*, *ERG11*, and *ERG25* were significantly upregulated, underscoring their pivotal role in ergosterol synthesis. Ergosterol, a crucial component of the yeast cell membrane, interacts with phospholipids to regulate membrane fluidity [30], permeability [31], the activity of membrane-bound enzymes, and substance transport, thereby stabilizing the cell membrane structure. Just as cholesterol is vital for rapidly dividing cells in mammals, ergosterol serves a similar function in *S. cerevisiae*. Exposure to exogenous substances such as bile acids leads to a marked upregulation of transport proteins, particularly ATP-dependent transporters like Pdr5p and Pdr16p, which increased 42.845-fold and 6.969-fold relative to control levels, respectively. *PDR5* is crucial for detoxification processes, as well as the transport of lipids, fatty acids, sterols, metabolites, and toxins [32]. The overexpression of PDR family transporters enhances yeast resistance to steroid metabolites, primarily through an extrusion pump mechanism that expels accumulated steroids or toxic metabolites from the cell. Research by Li et al. [33] further suggests that genetic disruption of the mitotic spindle assembly checkpoint leads to increased expression of PDR family efflux pumps and affects the function of PDR downregulation factors, thereby improving cellular tolerance to steroid metabolites. In summary, *S. cerevisiae* likely achieves bile acid tolerance by upregulating the expression of transport proteins. These transport proteins play a critical role in responding to exogenous steroid metabolites, maintaining cellular homeostasis, and supporting growth by regulating membrane permeability and metabolite expulsion.

To explore the interaction between bile acids and transport proteins in *S. cerevisiae*, the study utilized ergosterol as a control and compared the binding affinities of several bile acid molecules with these transport proteins. Figure 6a illustrates the first six entities as transport proteins, with Pdr3p serving as a transcriptional regulator. Molecular docking results revealed that cholesterol exhibited the highest binding affinity among the tested compounds. According to Seelig et al. [34], hydrogen bond acceptor groups, aromatic rings, and hydrophobic interactions are pivotal in the binding process between transport proteins and substrates. The docking results showed the following binding affinities: Pdr5p had an affinity of -9.5 kcal/mol with ergosterol and -7.9 kcal/mol with bile acid DCA; Pdr16p had an affinity of -7.5 kcal/mol with ergosterol and -7.5 kcal/mol with DCA; and Snq2p had an affinity of −8.8 kcal/mol with ergosterol and −9.2 kcal/mol with DCA. Specifically, Pdr5p interacts with ergosterol primarily through bonding forces, while its interaction with DCA involves four hydrogen bonds in addition to bonding forces. Pdr16p interacts with ergosterol through bonding forces and π-bonding, whereas its interaction with DCA involves bonding forces and hydrogen bonds. Snq2p, on the other hand, shows repulsive interactions with ergosterol and interacts with DCA predominantly through van der Waals forces and bonding forces. These findings suggest that Snq2p has a lower binding affinity for bile acids, indicating its primary role in bile acid transport. Conversely, Pdr5p shows a lower binding affinity for ergosterol, suggesting its principal function is in ergosterol transport. These results offer new insights into the distinct mechanisms through which bile acids and ergosterol interact with transport proteins.

Research indicates that deletion of the *PDR5* gene disrupts the ergosterol biosynthesis pathway, leading to ergosterol accumulation within cells and subsequent cellular toxicity [35]. Conversely, upregulation of *PDR5* enhances yeast cell detoxification. Miyahara et al. [36] demonstrated that high expression or hyperactive mutants of *PDR5* exhibit resistance to a diverse array of drugs, including cationic drugs, mutagens, antifungals, sterols, and anticancer agents. Additionally, Blazquez et al. emphasized that sterols are crucial activators of transport proteins, with bile acids playing a significant role in regulating ATP-binding cassette transporter G2 (ABCG2) activity [32,37,38]. The study results suggest that bile acids primarily enhance yeast tolerance to exogenous substances by stimulating the expression and activity of transport proteins such as Pdr5p, Pdr16p, and Snq2p. Ling et al. [39] reported that overexpression of the endogenous ABC transporter Snq2p significantly increases *S. cerevisiae*’s tolerance to exogenous decane and promotes strain growth. Watanabe et al. demonstrated that dual deletion of *PDR5* and *SNQ2* impairs yeast growth during the logarithmic phase. However, whether overexpression of these proteins can further enhance cell growth remains to be explored [39]. This highlights the crucial role of transport proteins like Pdr5p and Snq2p in regulating intracellular homeostasis and responding to exogenous stress. Nonetheless, their impact on cell growth and metabolism under varying conditions is not yet fully elucidated and warrants further investigation.

## 4. Conclusions

Research indicates that bile acids stimulate the overexpression of transport proteins, thereby enhancing the tolerance of *S. cerevisiae* to these steroid compounds. The inclusion of bile acids in the growth medium modulates transport proteins such as Pdr5p through the transcriptional regulator *PDR3*, thereby further increasing the yeast’s tolerance to bile acids and supporting its growth and development. Growth curve analyses suggest that bile acids may promote yeast growth; however, whether this effect is directly attributable to the overexpression of transport proteins in *S. cerevisiae* requires further experimental validation. Additionally, while ATP-binding cassette (ABC) transporters are known to influence drug resistance, their specific roles and mechanisms in yeast growth and development are yet to be fully elucidated. This study establishes a foundation for future research into the transformation of bile acids by *S. cerevisiae* and provides insights into the mechanisms underlying its resistance to these metabolites.

## Figures and Tables

**Figure 1 foods-13-03405-f001:**
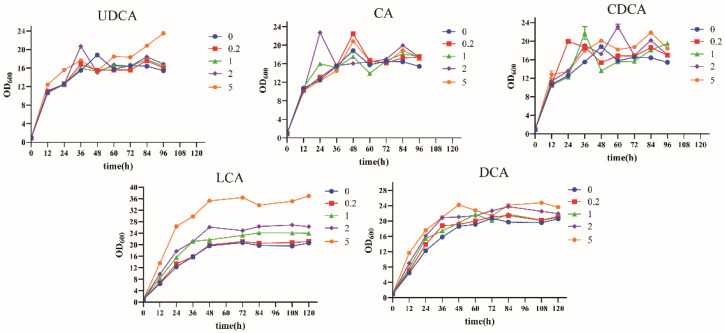
Growth curve of Saccharomyces cerevisiae. DCA: deoxycholic acid; CA: cholic acid; CDCA: chenodeoxycholic acid; LCA: lithocholic acid; UDCA: ursodeoxycholic acid. The concentrations of bile acids were: 0, 0.2, 1, 2, and 5 mg/mL.

**Figure 2 foods-13-03405-f002:**
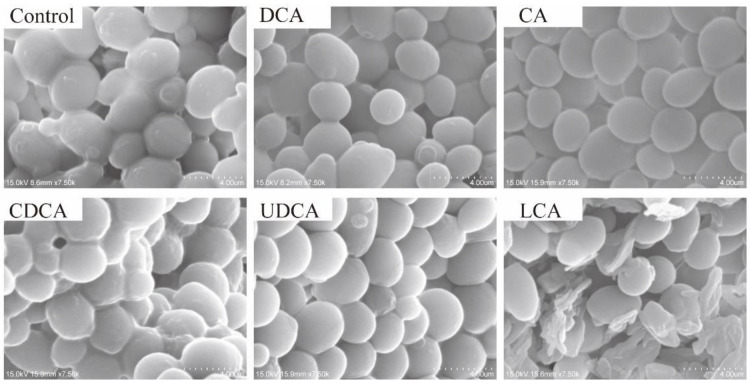
Morphology of *S. cerevisiae* cells under different bile acid cultures: magnification: ×75,000; scale bar: 4 μm. DCA: deoxycholic acid; CA: cholic acid; CDCA: chenodeoxycholic acid; LCA: lithocholic acid; UDCA: ursodeoxycholic acid.

**Figure 3 foods-13-03405-f003:**
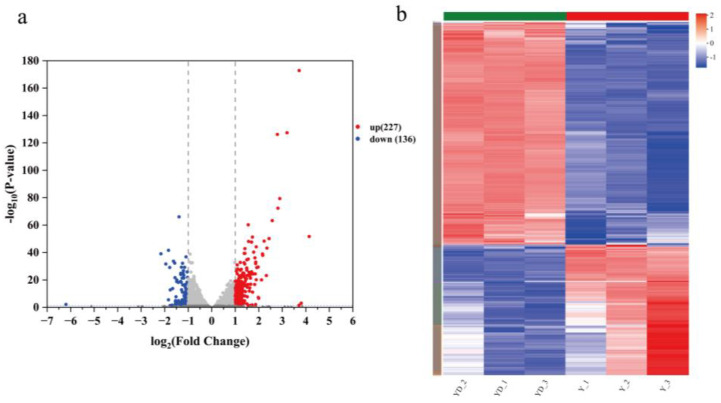
DEG and clustering analysis of significant gene changes after the addition of deoxycholic acid to *S. cerevisiae.* (**a**) Differentially expressed genes (DEGs) between Y and YD, with a fold change threshold of 2 and *p* < 0.05; (**b**) clustering analysis.

**Figure 4 foods-13-03405-f004:**
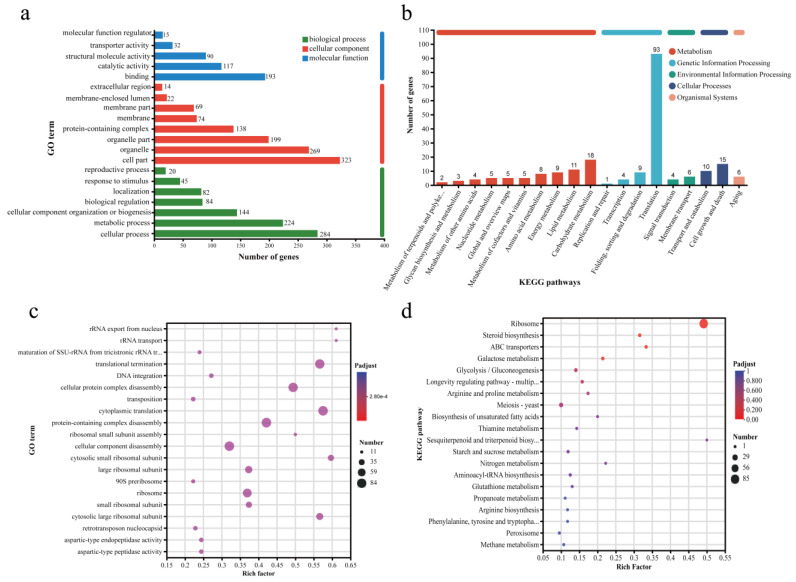
GO and KEGG enrichment analysis. The genes were classified and analyzed through Gene Ontology (GO) and Kyoto Encyclopedia of Genes Genomes (KEGG). (**a**) GO classification bar plot of gene sets, (**b**) KEGG classification bar plot of gene sets, (**c**) GO enrichment analysis bubble plot, (**d**) KEGG enrichment analysis bubble plot.

**Figure 5 foods-13-03405-f005:**
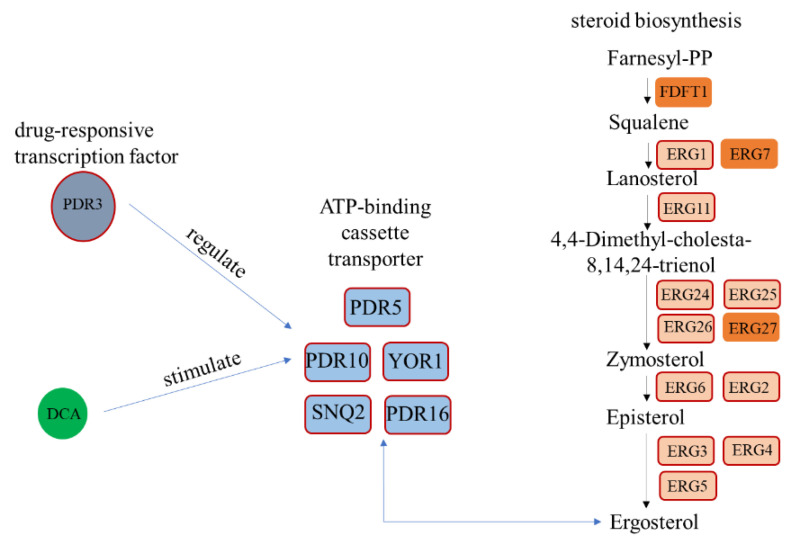
Mechanistic analysis of steroid regulation.

**Figure 6 foods-13-03405-f006:**
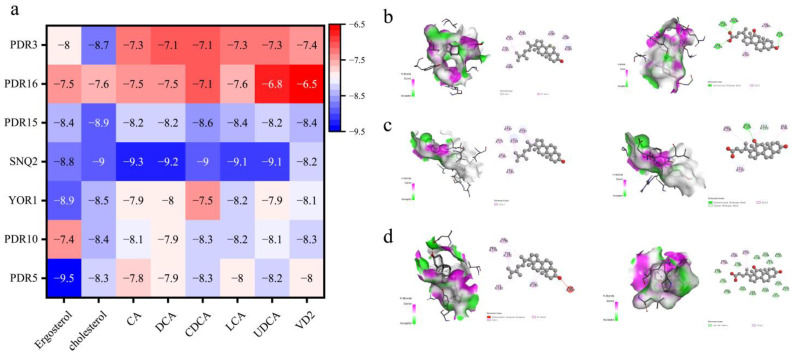
Docking of transport proteins with steroid molecules. (**a**) Docking results of transport proteins and regulatory factors with steroid metabolites. (**b**–**d**) Docking results of Pdr5p, Pdr16p, and Snq2p with ergosterol and DCA small molecules, respectively. The left side shows docking with ergosterol, and the right side shows docking with DCA. Docking was performed using AutoDock Vina (AutoDock Vina1.5.7, Trott et al., 2023), and visualization was carried out using Discovery Studio 2019 (Accelrys, 2019).

**Table 1 foods-13-03405-t001:** Expression of DEGs in ergosterol synthesis and transport pathways.

DEGs	Gene Description	YD/Y	Log2FC(YD/Y)	*p* Value
*PDR3*	drug-responsive transcription factor	1.765	0.819924	***
*PDR15*	ATP-binding cassette multidrug transporter	2.087	1.061711	***
*PDR16*	phosphatidylinositol transporter	6.918	2.79032	***
*PDR5*	ATP-binding cassette multidrug transporter	42.845	5.42104	***
*PDR10*	ATP-binding cassette multidrug transporter	2.958	1.564782	***
*YOR1*	ATP-binding cassette transporter	2.37	1.244614	***
*SNQ2*	ATP-binding cassette transporter	2.815	1.493139	***
*ERG26*	sterol-4-alpha-carboxylate 3-dehydrogenase decarboxylating	1.858	0.893546	***
*ERG4*	C-24 sterol reductase	2.002	1.001187	***
*ERG25*	methylsterol monooxygenase	2.172	1.119051	***
*ERG1*	squalene monooxygenase	2.142	1.099141	***
*ERG11*	sterol 14-demethylase	2.378	1.249492	***
*ERG3*	C-5 sterol desaturase	2.905	1.538689	***
*ERG6*	sterol 24-C-methyltransferase	1.911	0.934227	***
*ERG5*	C-22 sterol desaturase	3.71	1.891359	***
*ERG2*	C-8 sterol isomerase	1.606	0.683688	***

*** *p* < 0.001.

## Data Availability

The original contributions presented in this study are included in the article. Further inquiries can be directed to the corresponding author.

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
