# Peer review of "Elucidating Bile Acid Tolerance in Saccharomyces cerevisiae: Effects on Sterol Biosynthesis and Transport Protein Expression"

_foods, 2024, doi:10.3390/foods13213405_

Round 1
Reviewer 1 Report
Comments and Suggestions for Authors
The research activity reported in the manuscript focuses on elucidating the mechanisms of tolerance of a strain of S. cerevisiae to varying concentrations of bile salts, with a view to its potential use as a probiotic. While the article is of interest, it is nevertheless marred by numerous flaws in the text.
My suggestions are as follows:
Line 2: Saccharomyces cerevisiae in italics
Line 10: Saccharomyces cerevisiae in italics
Line 13: S. cerevisiae
Line 20: S. cerevisiae
Line 23: S. cerevisiae in italics
Line 35: S. cerevisiae in italics
Line 37: S. cerevisiae in italics
Line 41: S. cerevisiae in italics
Line 46: al.[6] successfully....
Line 53: S. cerevisiae in italics
Liens 58-60: Include at least one reference bibliography.
Line 74: Liu et al.[10]
Line 77: delete "[10]".
Line 89: S. cerevisiae
Line 102: S. cerevisiae
Line 103: 2 or 3?
Line 114: S. cerevisiae
Line 117: scanning electron microscopy (SEM) - Specify brand and mode (city and country)
Line 130: Nanodrop 2000: specify brand, city and country
Line 131: Agilent 5300 system - specify city and country
Line 146: Fastp - specify version, brand, city and country
Line 150: Hisat 2 - specify brand, city and country
Line 162: Origin 2021 software - specify brand, city and country
Line 163: GraphPad Prism 10 software - specify brand, city and country
Lines 164-166: Specify the software used for statistical analysis of the data
Lines 169-170: S. cerevisiae in italics
Line 173: "DCA、CDCA、UDCA" chech font (comma)
Lines 173-174: S. cerevisiae in italics
Line 183: Saccharomyces cerevisiae in italics; indicate in the figure legend the meaning of the acronyms UDCA, CA, CDCA, LCA and DCA.
Line 186: S. cerevisiae in italics
Line 196: S. cerevisiae in italics
Line 198: S. cerevisiae in italics
Line 201: "Figure 2" remove the italics; indicate in the figure legend the meaning of the acronyms UDCA, CA, CDCA, LCA and DCA; Saccharomyces cerevisiae in italics
Line 206: "Figure 3" remove the italics
Line 209: S. cerevisiae in italics
Line 220: "Figure 4" remove the italics
Line 224: S. cerevisiae in italics
Line 227: Figure 4a
Line 233: Figure 4b
Line 237: Figure 4c
Line 243: Figure 4d
Line 258: S. cerevisiae in italics
Line 271: S. cerevisiae in italics
Line 280: S. cerevisiae in italics
Lines 285-286: S. cerevisiae in italics
Line 298: "Figure 5" remove the italics
Line 299: indicate the figure number
Lines 304-305: S. cerevisiae in italics
Lines 325-326: S. cerevisiae in italics
Lines 337-338: S. cerevisiae in italics
Line 369: S. cerevisiae
Line 379: S. cerevisiae
Line 384: S. cerevisiae
Lines 388-389: S. cerevisiae
Line 401: Check the formatting of bibliographic references. Authors must follow the guidelines
Line 411: Saccharomyces cerevisiae in italics
Line 413: Saccharomyces cerevisiae in italics
Line 416: Saccharomyces cerevisiae in italics
Line 426: Saccharomyces cerevisiae in italics
Line 439: Saccharomyces cerevisiae in italics
Line 442: Saccharomyces cerevisiae in italics
Line 445: International Journal of Food Microbiology abbreviate as Int. J. Food Microbiol.
Line 452: Candida albicans in italics
Line 450: Saccharomyces cerevisiae in italics
Line 471: Saccharomyces cerevisiae in italics
Line 482: eliminate the double point
Author Response
Thank you for taking the time to read our manuscript. Your valuable comments have been explained. For details, please refer to the uploaded file and revised manuscript.

Reviewer 2 Report
Comments and Suggestions for Authors
The article addresses the effect of bile acids on growth, cell morphology and protein expression in Saccharomyces cerevisiae and focuses on altered expression of sterol biosynthesis proteins and transporters. The authors introduce the role of yeast in industry and in gut health. They then introduce bile acids in the human body and their role in digestion and effect on metabolism. Next the authors describe the role of yeast as probiotics and the effect of bile acids on this role.
The authors tested the effect of different bile acids (and different concentrations) on yeast growth in liquid culture. They go on to use scanning electron microscopy to study how bile acids affect the morphology of the yeast cell. Next they use transcriptomic sequencing to investigate changes in gene expression in S. cerevisiae in response to the presence of deoxycholic acid. The data sets of up- and downregulated genes were analysed using various software to establish which categories of genes are particularly affected by this bile acid.
The authors found that several bile acids promote the growth of yeast in liquid culture and discuss why they might have this effect. They also found that LCA has the greatest effect on the cell morphology of S. cerevisiae and relate this to previous findings by other groups. They identify 227 upregulated and 136 downregualted genes in DCA-treated, relative to untreated, yeast. Gene ontology anaysis revealed the differentially-expressed genes were enriched for those with roles in translation and the degradation of cellular protein complexes. KEGG pathway analysis showed that differentially-expressed genes were enriched for ribosomal processes, steroid synthesis, transport proteins and carbohydrate metabolism. Differentially expressed genes were also enriched for transport proteins including ABC transporters. The authors discuss how transporter expression and steroid biosynthesis could be differentially regulated in response to the presence of DCA. Finally, the authors used software to study the interaction between transporter proteins and different steroids. They relate low affinities with transporter activity and relate the results to the findings of other groups. The authors conclude that upregulated transporter expression facilitates rapid removal of bile acids and thus promotes growth in the presence of bile acids.
The paper is informative, logical and of interest to readers from a wide range of fields. The use of docking analysis to enhance the transciptomic results is particularly clever and greatly increases the relevance of the article. I would recommend the manuscript for publication but with the following changes:
The species name "Saccharomyces cerevisiae" should be italicised throughout the paper (eg. line 10)
Gene names should be italicised throughout the paper (e.g. 274: YOR1)
Protein names should be written correctly (e.g. 255: Pdr5p, Pdr10p, Pdr16p and Snq2p)
Figure legends should give a brief description of what was done
References to figures should include the full figure number and letter (e.g. 227: Figure 4a)
Software should be fully supported with version and citation (e.g. 150: Hisat2 2.2.1 Zhang et al, 2021)
11: probiotics should be a probiotic
30, 37: strain and strains should be species
40: de novo should be italicised
44: Se should be Sc
81: delete "the"
191/192: citation needed for "previous studies"
299: should be "Figure 5"
Table 1: PDR15, PDR5 and PDR10 need to be deleted from gene description column
333: Display should be display (lower case letter)
Attached PDF shows all places where correction necessary

Author Response

(The authors gave the same response as above.)

Reviewer 3 Report
Comments and Suggestions for Authors
Elucidating bile acid tolerance in Saccharomyces cerevisiae: Effects on sterol biosynthesis and transport protein expression is an interesting proposal for the article. After reading the article, some observations will be made.
In general, it is an interesting description of the article. Some recommendations are:
The writing of Saccharomyces cerevisiae in lines 10, 23, 35, 37, 41, 49, 53, 169, 174, 183, 186, 196, 198, 201, 209, 224, 245, 249, 258, 271, 280, 285, 286, 290, 411, 413, 416, 426, 439, 442, 450, 471, 478, it is incorrect, it should be italicized, and homogenize given that sometimes it writes Saccharomyces cerevisiae and sometimes S cerevisiae
The text accompanying a figure on line 182 must adequately describe it. Include information on concentrations evaluated and markers used for each condition. The text on line 182 alone does not sufficiently describe the figure; the same is true for the other figures.
In the results and discussion section, lines 167-271, there are no references. They are compared with previous studies on line 193, but no references exist.
Line 223 According to….. there is no reference
Line 256, no reference refers to the expression associated with enzymatic activity.
Line 261, thereby maintaining cellular, has no reference.
Line 262, Growth and adaptation, has no references.
It would be appropriate in this section to carry out an adequate discussion of the results based on the references provided.
Conclusion
is appropriate for the information presented
Author Response

(The authors gave the same response as above.)
